# Developing Gold Nanoparticles-Conjugated Aflatoxin B1 Antifungal Strips

**DOI:** 10.3390/ijms20246260

**Published:** 2019-12-12

**Authors:** Tobiloba Sojinrin, Kangze Liu, Kan Wang, Daxiang Cui, Hugh J. Byrne, James F. Curtin, Furong Tian

**Affiliations:** 1College of Sciences and Health, Technological University Dublin, Dublin 1, Ireland; Tobiloba.Sojinrin@tudublin.ie (T.S.); kangze.liu@tudublin.ie (K.L.); James.curtin@tudublin.ie (J.F.C.); 2Department of Instrument Science and Engineering, National Center for Translational Medicine, Shanghai Jiao Tong University, Shanghai 200240, China; wk_xa@163.com (K.W.); dxcui@sjtu.edu.cn (D.C.); 3FOCAS Research Institute, Technological University Dublin, Kevin Street, Dublin 8, Ireland; hugh.byrne@tudublin.ie

**Keywords:** gold nano particles, aflatoxin B1, lateral flow immunochromatographic strips

## Abstract

Lateral flow immunochromatographic assays are a powerful diagnostic tool for point-of-care tests, based on their simplicity, specificity, and sensitivity. In this study, a rapid and sensitive gold nanoparticle (AuNP) immunochromatographic strip is produced for detecting aflatoxin B1 (AFB1) in suspicious fungi-contaminated food samples. The 10 nm AuNPs were encompassed by bovine serum albumin (BSA) and AFB1 antibody. Thin-layer chromatography, gel electrophoresis and nuclear magnetic resonance spectroscopy were employed for analysing the chemical complexes. Various concentrations of AFB1 antigen (0–16 ng/mL) were tested with AFB1 antibody–BSA–AuNPs (conjugated AuNPs) and then analysed by scanning electron microscopy, ultraviolet–visible spectroscopy, and Zetasizer. The results showed that the AFB1 antibody was coupled to BSA by the *N*-hydroxysuccinimide ester method. The AuNPs application has the potential to contribute to AFB1 detection by monitoring a visible colour change from red to purple-blue, with a detection limit of 2 ng/mL in a 96-well plate. The lateral flow immunochromatographic strip tests are rapid, taking less than 10 min., and they have a detection capacity of 10 ng/g. The smartphone analysis of strips provided the results in 3 s, with a detection limit of 0.3 ng/g for AFB1 when the concentration was below 10 ng/g. Excellent agreement was found with AFB1 determination by high-performance liquid chromatography in the determination of AFB1 among 20 samples of peanuts, corn, rice, and bread.

## 1. Introduction

Aflatoxins (AFLs) are poisonous carcinogens and mutagens to both humans and animals that are produced by fungi [1]. Different commodities, such as corn, peanuts, and grains, have been frequently found to be contaminated with AFLs [2,3]. A positive correlation between AFLs contamination of agricultural commodities and primary hepatocellular carcinoma has been documented [2,3]. According to the World Health Organisation (WHO), AFLs destroy 25% of the world’s crops annually [4]. Not only are AFLs carcinogenic, but these mycotoxins have also been associated with growth impairment in children [5]. Currently, AFL contamination is a large problem for undeveloped areas of Africa and Asia, but it might soon include Europe and both the North and South America continents. A recent study published by *Nature* outlined that climate change might increase the levels of AFLs in European and American continental maize crops [6,7]. The South American trade bloc Mercosur has established permitted contamination limits for peanuts and peanut paste as 20 µg/kg for total AFLs and 5 µg/kg for Aflatoixin B1 (AFB1) [8]. The European Economic Community (EEC) has established permitted food contamination limits of 2 µg/kg for AFB1 and 4 µg/kg for the total concentration of the four AFLs since 1 February 1999 [9]. Therefore, it is necessary to develop strategies for achieving the limits of AFL contamination and reducing AFL exposure in vulnerable populations [10].

Thin-layer chromatography (TLC) and high-performance liquid chromatography (HPLC) are the most popular techniques for detecting AFLs. However, these methods require extensive sample preparation, expensive instruments, and operation by skilled professionals. Alternatively, the enzyme-linked immunosorbent assay (ELISA) has been successfully developed for AFLs [11], but ELISA also needs incubation and washing steps, and application is mainly confined to laboratories. Lateral flow immunochromatographic/immunoassay strips (LFIAs) have received increasing attention for qualitative and quantitative analysis in different scientific sectors [12], including food safety, environmental monitoring, and precision medicine [12,13]. In 2005, Delmulle et al. [14] developed an LFIA for the detection of aflatoxin B1 (AFB1) in pig feed. Liao and Li [15] have made significant effort to investigate the effect of the core–shell silver–gold nanocomposites on the properties of LFIAs. However, this detection can only provide either qualitative (positive or negative) or semi-quantitative information on analyte concentration, and thereby does not satisfy the requirements for practical applications [8,16]. Moreover, Anfossi et al. developed a quantitative LFIA for the detection of aflatoxins in maize [17]. A competitive reaction between a biotin-modified aptamer specific to AFB1 and fluorescent cyanine 5-modified DNA probes formed the basis of a dot assay that Shim et al. developed on an LFIA test strip for detection of AFB1 [18]. A fluorescence detection apparatus that was coupled to a desktop computer or laptop, enabling rapid processing speeds and stable performances, recorded the fluorescence intensity of the dot. However, these bulky and heavy devices limit their widespread application in the field of family and personal care [19,20,21]. Alternatively, a mobile device-based strip reader could satisfy the requirement of high portability and feature-rich testing. The mobile health market is rapidly developing, and portable diagnostic tools provide an opportunity to increase the accessibility of health care and decrease costs [22]. Following the developments of various smartphone-based strip readers for quantitative measurements of human diseases [23,24,25,26,27,28,29,30,31], smartphone analysis for the detection of AFL on LFIAs has been also reported earlier this year [32]. The limit of detection (LOD) of gold nano particles (AuNPs) based LFIA has been dramatically improved from 10 μg/mL to 1 ng/g [1,2,3,8,14,18]. This scenario motivated the development of new strategy providing quantitative analyte concentration for testing LFIAs. So far, AuNPs that are sized 30-40 nm for AFB1 conjugation have been reported in literatures [1,2,3,8,14,18]. Di Nardo et al. have employed blue (desert rose-like, mean diameter ca.75 nm) AuNPs in order to produce different colour bands of LFIAs [32,33]. There is a strong association between the AuNPs formulation and colour change [34,35]. The associated colour can be employed for a number of applications and, therefore, continued refinement of AuNPs synthesis can provide desirable bands for LFIAs.

This study aims to develop a small gold nanoparticle (AuNP) immunochromatographic strip for detecting AFB1 in food samples. Firstly, 10 nm AuNPs will be encompassed by bovine serum albumin (BSA) and AFB1 antibody to form anti-AFB1 antibody–BSA nano complexes. Afterwards, nuclear magnetic resonance (NMR) spectroscopy, thin-layer chromatography (TLC), gel electrophoresis, and scanning electron microscopy (SEM) will be used to characterise the chemical complexes of AuNPs, BSA, and AuNP with AFB1 antibody–BSA. The colour change of the complex with different concentrations of AFB1 will be quantified according to the spectroscopic signature of the surface plasmon resonance (SPR) in a 96-well plate. The complex will be employed in a LFIA to further elucidate the advantage of 10 nm AuNPs. The density of the test line (T-line) and control line (C-line) will be analysed by visual and smartphone-based imaging systems. Additionally, a portable smartphone strip reader with grey-scale processing, improved Sobel convolution operator, threshold analysis, and image binarization will be employed to analyse the strips. AFB1 in peanuts, corn, rice, and bread will be determined by the immunochromatographic strip and for comparison.

## 2. Results

### 2.1. H-NMR Spectra of AuNP Synthesis

The signal in the conjugate spectrum was a three-proton singlet at 3.95 ppm, being identified as the aromatic *O*-methyl resonance of the AFB1 antibody moiety (H-g, Table 1, Figure 1 and Figure 2). In comparison to the conjugate spectrum, AFB1 antibody showed the absence of olefinic coupling in the conjugate. Two singlets at 4.05 ppm and 4.22 ppm were identified as the resulting aliphatic protons after addition (H-h and H-i, Table 1, Figure 1 and Figure 2). Two doublets at location 9 and location 16 were assigned to the *cis*-protons (H-k and H-l, Table 1, Figure 1 and Figure 2) of the two furan ring systems, and decoupling confirmed the assignment. A proton singlet at d 6.52 ppm was identified as the aromatic proton (H-m, Table 1, Figure 1 and Figure 2). The citrate-stabilised AuNPs carried a net negative surface charge and they were immediately bound with BSA *via* ionic bonds to form a stable AFB1 antibody–BSA–AuNPs conjugate (conjugated AuNPs, Appendix A).

### 2.2. Confirmation of Physical Characteristics of AuNPs

SEM and SPR were carried out on pristine AuNPs, conjugated AFB1 antibody–BSA–AuNPs (conjugated AuNPs) and conjugated AFB1 antibody–BSA–AuNPs that were mixed with 2 ng/mL antigen (conjugated + antigen, Figure 3a,b). The blue curve in Figure 3g indicates that the SPR of plain AuNPs had a peak at 520 nm wavelength (Figure 3g). The orange curve denoting the SPR of conjugated AuNPs indicates an increased absorbance, and the peak slightly shifted to 530 nm. The grey line represents the SPR of conjugated AuNPs with 2 ng/mL antigen (Figure 3g). Figure 3d–f depict the particle sizes of plain NPs, conjugated AuNPs, and conjugated AuNPs + antigen.

The SPR at 520 nm wavelength is consistent with spherical pristine AuNPs with a diameter of 10 nm (Figure 3a) (blue curve in Figure 3g). The conjugated AuNPs slightly increased in size, due to conjugation, which translated into a shift in the SPR peak. Figure 3e visually illustrates the increase in size, which depicts an SEM image of the conjugated AuNPs. The average particle size was 14 ± 2 nm (Figure 3f, zeta-potential of 30 ± 2.8 mV). Particle aggregation occurred when the antigen was mixed with the conjugated AuNPs. During the mixing process, the colour of the wells changed from pink to purple/blue and then grey. The colour changes of the colloidal gold were sufficient to be observed by the naked eye. The time from the start of mixing to the time at which the colour changed to purple/blue was recorded (Figure 3h). In the first ~10 min., the size of the AuNPs increased and remained relatively stable for the following ~100 min. The colour changed from red to purple-blue. The SPR became relatively flat in shape and low in absorbance at 620 nm after 10 min. (green line in Figure 3e). The particle size was 50 ± 9 nm (zeta-potential of 30 ± 3.9 mV). Aggregation started at~120 min. after mixing, as the particle size gradually increased. The particle size was 92 ± 11 nm (zeta-potential of 37 ± 4 mV) at 4 h. The particles size reached to 860 ± 32 nm (zeta-potential of 42 ± 4.6 mV) at 16 h. The data showed that the particle size was 1350 ± 38 nm (zeta-potential of 50 ± 4 mV) at 24 h. (Figure 3h).

### 2.3. AFB1 antibody–BSA–AuNPs in 96-well Plate

Various concentrations of AFB1 (0, 1, 2, 4, 8, and 16 ng/mL) were used to optimise the sensitivity of AFB1 antibody–BSA–AuNPs with the antigen (Figure 4a). The absorbance ratio between the non-aggregated/negative sample (at 520 nm) and aggregated AuNPs (at 620 nm) was correlated with the colour change at 10 min. after mixing. The ratio of absorbance at 520/620 nm was employed as an indicator of the process. The absorbance ratio was calculated for each sample, and the mean values were plotted against the concentrations of AFB1. The absorbance ratio exhibited a plateau below ~2 ng/mL when the concentration varied from 0–1 ng/mL. When the concentration was higher than 2 ng/mL, the absorbance ratio increased and reached a maximal peak, then decreased afterwards. The absorbance ratio changed at 2 ng/mL, which defined the limit of detectability, with an approximately linear region between 2–16 ng/mL (Figure 4c).

### 2.4. Comparison between Strips and HPLC for AFB1 Detection in Food Samples

A range of concentrations of AFB1 antigen (0–50 ng/mL) were analysed by strips and HPLC (Figure 5, Figure 6 and Figure 7, Appendix A). The correlation between the peak area and concentration of AFB1 antigen had a coefficient *R*^2^ of 0.9929 (Figure 5e).

The operation of the strips is based on a competitive reaction between the free AFB1 contained in the sample and the fixed coating of antigen applied onto the nitrocellulose membrane. The sample solution that was added to the sample pad migrated towards the absorption pad, due to capillary forces. When the sample was positive, the conjugated antibody AFB1-BSA-AuNPs bonded to free AFB1 first *via* diffusion, resulting in a lower amount of conjugated antibody AFB1-BSA-AuNPs triggered by the AFB1 fixed on the T-line, leading to a less intense T-line. When the AFB1 concentration in the sample increased, the purple T-line disappeared. In contrast, the conjugated AuNPs bound to coating antigen AFB1 and goat anti-mouse IgG when the sample was negative, which resulted in a visible T-line and C-line (Figure 6). The C-line consistently emerged, indicating the effective assembly of the strips. The results were observed at 10 min. after loading the samples on the sample pads.

A series of concentrations of AFB1 antigen were analysed while using the strip. As shown in Figure 7a, the line intensity decreased when the concentration of AFB1 increased. At 10 ng/mL, AFB1 generated a distinct visible difference in the colour intensity of the T-line between the sample and the blank (no AFB1). The LOD by the naked eye was 10 ng/mL. The colour changes in the T-lines at different concentrations were not obvious to the naked eye (Figure 7a), but the intensity of the lines could be more sensitively recorded with a smartphone reader. After recording the intensity of the lines, a standard curve was obtained by plotting the linearity of T/C against the concentration of AFB1. The result of the smartphone analysis is shown as the correlation between the ratio of the T-line density/C-line density *versus* the concentration (Figure 7b). There was a four-parameter logistic relationship between the ratio of T/C density and AFB1 concentration (a = 0.1, d = 10, c = 0.98, b = 5.1 in equation). The correlation coefficient (*R*^2^) was relatively high (Figure 7b). 

AFB1 was extracted by methanol from food samples (peanut, bread, corn, and rice) to evaluate the possible application of the LFIAs for food screening, due to its hydrophobic property and the extracts assayed in triplicate while using the LFIAs and HPLC. For a positive sample that was identified by the LFIAs, there was only a purple line at the C-line of the membrane. For a negative sample, there was a visible purple line at the T-line and C-line of the membrane (Appendix A). When the sample strips showed negative results, the strips were inserted in the smartphone reader (Appendix A). The LFIAs (visual/smartphone) were evaluated against the existing HPLC method on the same samples (Table 2). The lowest amount of analyte in a sample that can be detected was 0.31 ng/g from corn (Table 2). Notably, the LFIA method was much simpler and faster (10 min), and the ideal detection limit of the system was found to be 0.3 ng/g.

## 3. Discussion

### 3.1. Confirming Conjugation of AuNPs Using TLC, Gel Electrophoresis and NMR

The reaction product was analysed by TLC, gel electrophoresis, and NMR in order to confirm the coupling of AFB1 antibody to AuNPs. The TLC results showed that the spots of the conjugated AuNPs did not move in the chloroform–acetone (9:1, *v*/*v*) solvent system, while the AFB1 antibody reached close to the solvent front. The spot of AFB1 antibody migrated faster than the one of AFB1 antibody conjugated AuNPs up the TLC plate. The conjugated antibody AFB1-BSA-AuNPs presented less polar, because the conjugation neutralized the electronegativity of the antibody. For the gel electrophoresis, the AuNPs complexes were separated by their different overall electric charges and sizes (i.e., plain, conjugate–antigen, and conjugate present on the electrophoresis gel, Appendix A). The plain AuNPs moved due to their negative charge in citrate solution. The conjugated antibody AFB1-BSA-AuNPs moved comparatively faster due to the negative charge of the antibody. The conjugated antibody AFB1-BSA-AuNPs mixed antigen was trapped in the electrophoresis gel and formed a purple band. The *N*-hydroxysuccinimide (NHS) ester method coupled the AFB1 antibody to BSA and AuNPs. These chemical-shift at H-h and H-i were consistent with substitution adjacent to the two protons, as demonstrated by the ^1^H-NMR spectra of AFB1 antibody–BSA [36]. The total absence of coupling suggested *trans*-configuration. The NMR results showed that the AFB1 antibody and BSA were successfully conjugated to the AuNPs (Figure 1 and Figure 2, Table 1). The most established conjugation of AFB1 to a protein is at the 1-position while using a carboxymethyloxime spacer [37]. However, this position has been taken by the anti-mouse antibody (Figure 2). This scenario requests a position opposite to the two furan rings of AFB1 to avoid hydrolysis sensitivity of the resulting conjugates and/or partial destruction of the toxin structure during the conjugation procedure [36].

### 3.2. Define the LOD of Conjugated AuNPs in 96-Well Plate

The ratios of SPR were employed to determine the LOD of conjugated AuNPs in the 96-well plate (Figure 4). The changes in the AuNP sizes are accompanied by a shifting of the SPR. The characteristics of the SPR depend on the nanoparticle size, which can strongly influence the resonant frequency that determines the characteristic colour, which is observable by the naked eye [34]. The associated colour can be employed for fungal detection [35]. The colour changes from red to purple/blue are sufficient for detection by the naked eye. The lowest concentration was at 2 ng/mL, when the colour changes were visible. The LOD in the 96-well plate assay was 2 ng/mL.

### 3.3. Comparison between HPLC and Immunochromatographic Strip for the Detection of AFLs

The concentration of AFB1 that was detected by the strips agreed with the HPLC analysis (Table 2). It has been reported that the results of ELISA and HPLC showed high correlation for 28 peanut samples [38]. The LOD was 5 µg/kg. However, the HPLC equipment is rather intimidating to new users. A HPLC apparatus can cost 20,000 to 0.5 million Euros. It also requires expensive running reagents, a specific column, and a trained practitioner to operate the equipment. The running time of the HPLC is 30 min. per sample. The similarity of outcome between the smartphone LFIAs and HPLC holds significant promise for rapid (10 min.) and accurate LFIAs, with the advantages of low-cost (1 Euro per strip) as compared to HPLC.

Researchers have developed several methods for the quantitative analysis of AFLs to detect the new limit of AFL contamination [39,40]. Table 3 shows the comparison of the materials, sensitivity, rapidity, and simplicity in the LFIAs based on aptamer or antibody. It has been reported that AFB1 has been conjugated with BSA and AuNPs in designing a one-dot assay on an LFIA test strip for a detection limit of 10 µg/mL AFB1 [41]. Furthermore, Delmulle et al. [14] developed an immunoassay-based lateral flow dipstick for the detection of AFB1 in pig feed, with a visual detection limit of 5 µg/kg. Moreover, the lateral flow dipstick that was developed by Liao and Li [15] using core–shell silver–gold nanocomposites attained a visual detection limit of 0.1 ng/mL AFLB1. However, the silver ion is toxic to zebrafish, and bacterial and mouse embryonic stem cells [42,43,44]. The AuNPs are stable, and spherical AuNPs have low toxicity to human cells and fungi [45,46]. As mentioned above, the 30-min. aptamer-based dipstick assay for fluorometric detection of AFB1 is another alternative, as developed by Shim et al. [18]. Di Nardo et al. has employed 75 nm AuNPs for the blue colour bands [32]. In comparison to Liao and Li’s method [15], we have produced environmentally-friendly strips while using 10 nm AuNPs conjugated with AFB1 antibody and BSA. The colour changes can be observed from red to purple/blue when the complex is mixed with over 2 ng/mL AFB1. The visual detection limit of the LFIAs is 10 ng/g. Additionally, we employed a portable smartphone strip reader with multiple image processing to analyse strips in 3 s [19,21]. The LOD is 0.3 ng/g. These limits are low enough to comply with the limits of the current legislation of 5 µg/kg AFB1. The results demonstrated that the LFIAs that were developed in this study could be useful as a quantitative and qualitative method for AFB1 detection in real samples.

## 4. Materials and Methods

### 4.1. Material

Gold colloid suspensions with a particle size of 10 nm (6 × 10^12^/mL), Bovine serum albumin (BSA), 1-ethyl-3-(3-dimethylaminopropyl)carbodiimide (EDC), dicyclohexylcarbodiimide, Trifluoroacetic acid (TFA), N-hydroxysuccinimide (NHS), AFB1, and anhydrous dimethyl sulphoxide (DMSO) were purchased from Sigma–Aldrich (Arklow, Co. Wicklow, Ireland). AFB The monoclonal antibody (6A10) was purchased from Thermo Fisher Scientific (Dublin, Ireland). Ultrapure deionised water (resistivity greater than 18.0 MΩ/cm) was used for all of the solution preparations and experiments.

### 4.2. Synthesis of AFB1-conjugated AuNPs

Dry glycolic acid (1 g) was dissolved in dry TFA (4 mL). AFLB1 antibody (10 mg) was dissolved in dry acetonitrile (4 mL) directly in the septum covered vial that was obtained from the distributor. This solution was added to the glycolic acid/TFA mixture while using a syringe at room temperature while stirring. The 54 µg of AFLB1 conjugates with 80 µL of *N,N*-dimethylformamide (DMF) in phosphate buffer at pH 5.7 (6:9, *v/v*), 8 µL of 0.1 M EDC, and 8 µL of 0.1 M NHS were added, followed by incubation for 15 min. The solution was then added drop-wise to BSA solution (1 mg/mL in carbonate buffer, pH 9.5) and then incubated at 25 °C for 2 h. The AFLB1 antibody–BSA solution was diluted with a solution of 0.05 mM phosphate-buffered saline (PBS, pH 7.2). Next, 1.0 mL of AFLB1 antibody–BSA mixture was added to 1.0 mL of AuNP solution, shaken for 2–3 s to mix, and then sonicated for 30 min. at room temperature. The solution was centrifuged at 3500 rpm to remove excess BSA and then re-suspended in water. This step was repeated at least three times, and the samples observed while using a spectrophotometer to ensure the excess BSA had been removed. After freeze-drying, the purified complex was dissolved in DMSO for NMR analysis.

### 4.3. NMR

The BSA, AFB1 antibody, and conjugated AuNPs were placed in a dialysis tube and then clipped at both ends. The samples were dialysed against 0.1 M sodium bicarbonate NaHCO_3_ for 12 h. A few drops of the conjugated AuNPs were added using a disposable pipette and dissolved in 1 mL of deuterated Dimethyl Sulfoxide (DMSO) and then transferred to an NMR tube for analysis. The samples were transferred to an NMR tube for analysis while using a Bruker Ascend 500 MHz spectrometer and deuterated DMSO-d6.

### 4.4. Thin layer Chromatography (TLC)

TLC was performed by spotting the conjugated AuNPs on Lane 1 and the AFB1 antibody standard on Lane 2 at a position that was 1 cm from the base of the silica gel plate, followed by running the TLC plate in a mixture of chloroform and acetone (9:1, *v/v*). The result was visualised under UV exposure.

### 4.5. Characterisation of AuNPs

A Perkin Elmer Lambda 900 UV/VIS/NIR spectrometer was used to measure the absorbance and observe the formation and stability of the AuNPs. A Zetasizer Nano ZS analyser (Malvern Instruments, Worcestershire, UK) was used to measure the hydrodynamic particle size and zeta-potential of the nanoparticles. Six replicates were measured for each data point. For gel electrophoresis, an aqueous solution of 0.8% (*w/v*) agarose gel was prepared and immersed in Tris–borate–EDTA buffer. The gel of 7 × 7 cm size and 1 cm thickness was run on a horizontal electrophoresis system with an electrode spacing of 15 cm (Mini-Sub Cell GT System, Bio-Rad, Hercules, CA, USA). After electrophoresis at 200 V, the AuNPs were separated according to their different electric charge (i.e., plain AuNPs, conjugated AuNPs, conjugated AuNPs + antigen), and the gel stained for the visualisation of the bands.

### 4.6. SEM of AuNPs

The pristine AuNPs, conjugated AuNPs and conjugated AuNPs + antigen mixed for 10 min. were deposited onto silicon substrates and characterised by SEM while using a Hitachi SU6600 FESEM instrument at an acceleration voltage of 25 kV. The SEM images were taken using the secondary electron detector.

### 4.7. Layout of AuNPs and Concentration of Antigen in 96-well Plates

The pristine AuNPs and conjugated AuNPs were added to the 96-well plate. The wells of the first row were the blanks containing AuNPs with 100 μL of ultrapure deionised water. The wells in rows 2–5 were filled with 100 μL of plain AuNPs suspensions, AuNP with conjugated anti-AFL. The concentrations of antigen were 0, 1, 2, 4, 8, and 16 ng/mL (well numbers 5–10) from left to right in each row. The wells in rows 2–5 were replicas for each 96-well plate. Four replicates were measured for each data point for SPR analysis of AuNPs.

### 4.8. HPLC

A total of 20 samples of peanuts, puffed corn, puffed rice, and bread were collected in a sterile container at room temperature. The same day, the samples were extracted, and the extracts were tested for AFB1 while using the LFIAs and HPLC. Briefly, 20 g of samples, including peanuts, puffed corn snack, puffed rice snack, and bread, was shaken with 5 g NaCl and 300 mL of methanol/water (80:20, *v/v*) for 30 min. For peanut and puffed corn snack, 100 mL *n*-hexane was also added. After filtration, 20 mL of the filtrate was diluted with 130 mL of deionised water and filtered through a glass microfiber filter, and 75 mL of the filtrate was used for further clean-up on an AFB1 test column. The samples were run on a HPLC instrument that was equipped with a C18 column and a PDA detector set at excitation and emission wavelengths of 210 and 435 nm, respectively. The mobile phase was water/methanol (55:45, *v/v*) with a flow rate of 1.2 mL/min.

## 5. Conclusions

In summary, the NHS ester method coupled the AFB1 antibody to BSA, and BSA was bound to AuNPs *via* ionic forces. The novel AuNPs-conjugated AFB1 antibody derivatives were used in developing a colorimetric assay in 96-well plates and the LFIA strips. The AuNPs application has the potential to contribute towards AFB1 detection by monitoring a colour change from red to purple/blue with the naked eye, and the LOD of the 96-well plate assay is 2 ng/mL. The visual detection limit of the LFIAs is 10 ng/g. Excellent agreement was found when compared with AFB1 determination by HPLC in the determination of AFB1 in 20 samples of peanuts, corn, rice, and bread. In this study, we report the smartphone-based LFIAs for AFB1 detection in food samples. The LOD is 0.3 ng/g.

## Figures and Tables

**Figure 1 ijms-20-06260-f001:**
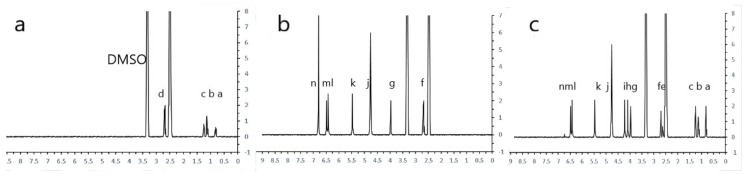
^1^H-nuclear magnetic resonance (^1^H-NMR) spectra of gold nanoparticle synthesis. ^1^H-NMR spectra of (**a**) bovine serum albumin (BSA), (**b**) Aflatoixin B1 (AFB1) antibody, (**c**) AFB1 antibody + BSA + gold nano particles (AuNPs) (Conjugated AuNPs). The peaks at 3.5 and 2.5 ppm indicate dimethyl sulphoxide (DMSO) and H_2_O, respectively. The lowercase letters (a–n) are listed in the first column of Table 1. The chemical shifts of protons are shown in the Table 1.

**Figure 2 ijms-20-06260-f002:**
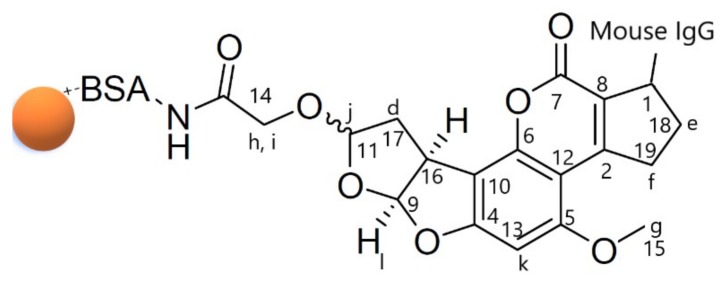
Structure of AFB1 antibody + BSA + AuNPs. The lowercase letters (a–n) are listed in the first column of Table 1. The chemical shifts of protons are shown in the fourth column of Table 1.

**Figure 3 ijms-20-06260-f003:**
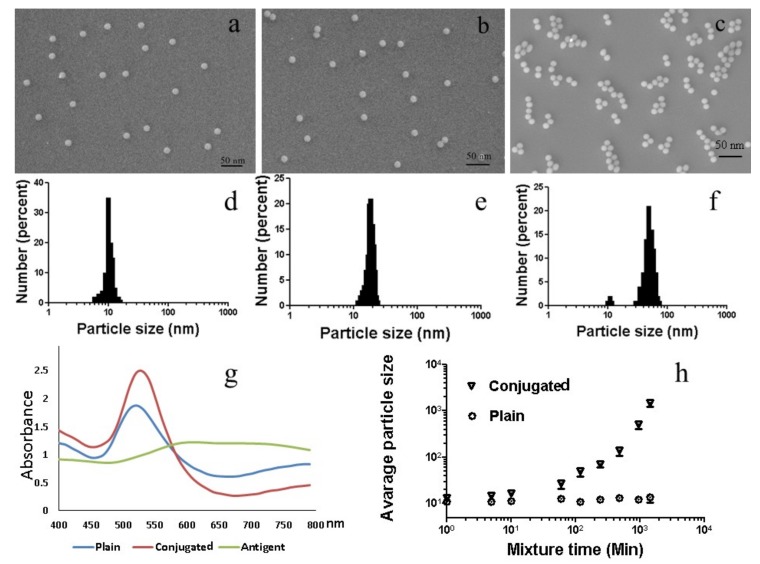
Gold nanoparticle characterisation. (**a–c**) The scanning electron microscopy (SEM) images of gold nanoparticles: (**a**) Plain, (**b**) Conjugated, (**c**) Conjugated + antigen. (**d–f**) Particle sizes distribution of (**d**) Plain, (**e**) Conjugated, (**f**) Conjugated +antigen. (**g**) Surface plasma resonances of gold nanoparticles, in which the blue curve shows plain AuNPs, the orange curve shows the conjugated AuNPs, the grey curve shows the conjugated AuNPs+ antigen. (**h**) The particle size change of the plain AuNPs and conjugated AuNPs mixing with 2 ng/mL antigen as a function of time.

**Figure 4 ijms-20-06260-f004:**
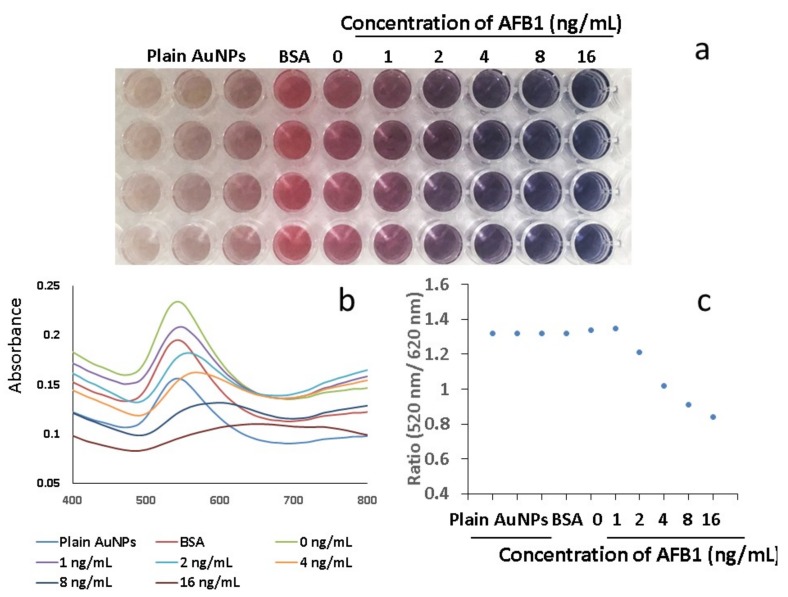
Surface plasmon resonance (SPR) of conjugated AuNPs reacted with antigen in 96-well plate (*n* = 4). (**a**) The different concentrations of AFB1 against conjugate AuNPs in 96-well plate. The wells 1–3 show plain AuNPs, and well 4 shows conjugated AuNPs without BSA. The concentration of AFB1 increases from well 5 to well 11 (from left to right) with a concentration of 0, 1, 2, 4, 8, and 16 ng/mL respectively. The wells from first row to fourth row are replicates. The conjugated AuNPs with BSA appear light pink (well 4). The conjugated AuNPs without antigen show red (well 5). The conjugated AuNPs mixed with antigen at concentration of 1 ng/mL show pink/purple (well 6). The conjugated AuNPs show purple colour when the concentration of antigen increases to 2 ng/mL. (**b**) Spectra of the conjugated AuNPs react with antigen. (**c**) Ratio of nanoparticle spectra absorbance at 520 nm/620 nm. The absorbance ratio 520 nm/620 nm gradually increases after the conjugated AuNPs are mixed with antigen. The ratio 520 nm/620 nm decreases when it reaches 0.8. The ratio of absorbance exhibits a plateau below 0–1 ng/mL and above 2 ng/mL, which defines the limits of detectability, and an approximately linear region in between 2 ng/mL to 16 ng/mL.

**Figure 5 ijms-20-06260-f005:**
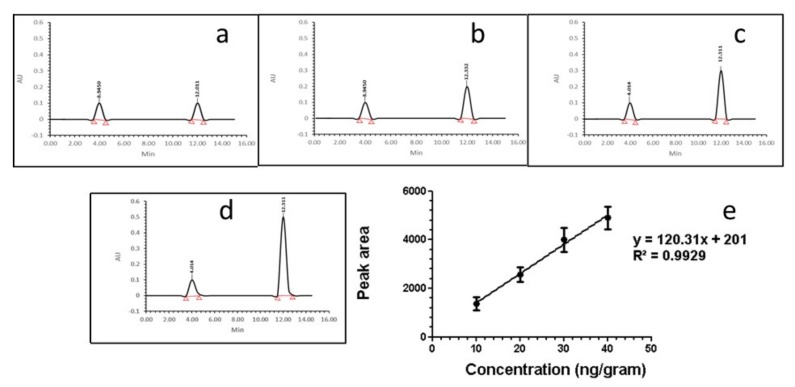
Standard curve of HPLC. (**a–d**) HPLC of different concentrations of AFB1. The peak around 12 min. indicates the AFB1; and, (**e**) A standard curve of concentration of AFB1 and peak area of HPLC at 12 min.

**Figure 6 ijms-20-06260-f006:**
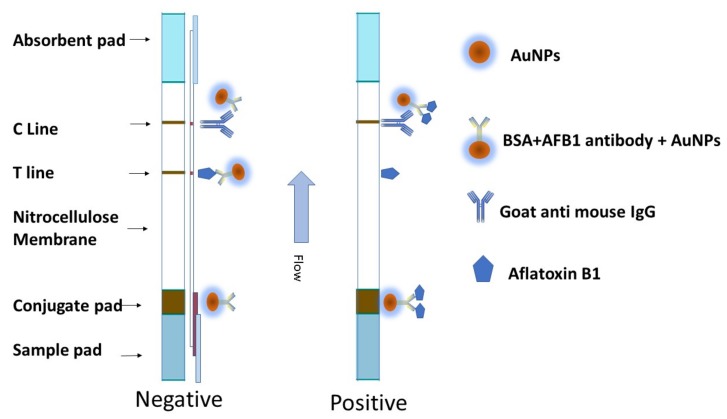
The scheme of gold nanoparticles based immunochromatographic strip.

**Figure 7 ijms-20-06260-f007:**
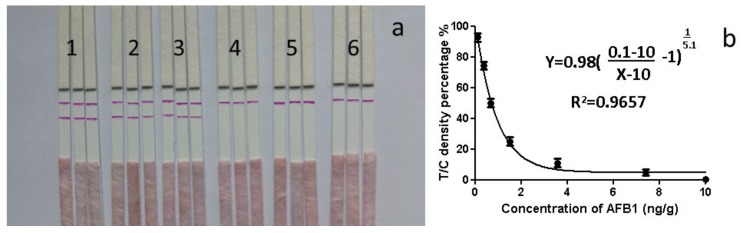
Standardisation of strips. (**a**) Images of strips, The C line is top line and T line is bottom line. 1 = 0 ng/mL; 2 = 2 ng/mL, 3 = 5 ng/mL, 4 = 10 ng/mL, 5 = 25 ng/mL, and 6 = 50 ng/mL, *n* = 6. (**b**) Normalisation of concentration of AFB1 and Ratio of density of T line/C line.

**Table 1 ijms-20-06260-t001:** ^1^H-NMR spectra of bovine serum albumin (BSA), Aflatoxin B1 (AFB1), and proposed AFB1 Antibody-BSA-AuNPs conjugate.

^1^H	BSA	AFB1 Antibody	AFB1 Antibody–BSA–AuNPs ppm
a	0.8 (J = 0.6 Hz)		0.8 (J = 2 Hz)
b	1.1 (J = 1.2 Hz)		1.1 (J = 1.2 Hz)
c	1.25 (J = 0.8 Hz)		1.25 (J = 2 Hz)
d	2.55 (J = 2 Hz)		2.55 (J = 0.2 Hz)
e		2.69 (J = 2 Hz)	2.64 (J = 2.1 Hz)
f		3.34 (J = 2 Hz)	3.34 (J = 2 Hz)
g		3.97 (J = 6 Hz)	3.95 (J = 6 Hz)
h			4.05 (J = 2 Hz)
i			4.22 (J = 2 Hz)
j		4.76 (J = 2.1 Hz)	4.76 (J = 2.1 Hz)
k		5.47 (J = 2.4 Hz)	5.47 (J = 2 Hz)
l		6.42 (J = 2.7 Hz)	6.33 (J = 2.4 Hz)
m		6.49 (J = 2 Hz)	6.52 (J = 2.1 Hz)
n		6.80 (J = 2.3 Hz)	

**Table 2 ijms-20-06260-t002:** Comparison between the visual/smart phone analysis lateral flow immunochromatographic assay strips (LFIAs) and the high-performance liquid chromatography (HPLC) method.

Sample	LFIAs		HPLC (ng/g)
	Strip Visual (+,−)	Smartphone (ng/g)	
Peanut	−,−,−	5.38 ± 0.05	5.70 ± 0.55
	−,−,−	4.38 ± 0.15	4.07 ± 0.83
	−,−,−	5.17 ± 0.12	5.05 ± 1.19
	+,+,+	Over10	42.53 ± 7.76
	−,−,−	5.18 ± 0.04	5.28 ± 1.36
Bread	−,−,−	2.03 ± 0.44	2.02 ± 0.56
	−,−,−	5.18 ± 0.04	5.21 ± 0.23
	+,+,+	Over10	27.67 ± 6.43
	−,−,−	4.01 ± 0.30	4.05 ± 0.61
	−,−,−	5.59 ± 0.06	5.71 ± 0.55
Corn	+,+,+	Over10	19.16 ± 4.25
	−,−,−	0.37 ± 0.01	0.31 ± 0.15
	−,−,−	2.27 ± 0.97	2.50 ± 0.54
	−,−,−	4.79 ± 0.29	5.03 ± 0.66
	−,−,−	5.38 ± 0.53	5.32 ± 0.99
Rice	−,−,−	2.94 ± 0.81	3.00 ± 1.15
	−,−,−	0.42 ± 0.09	0.36 ± 0.20
	−,−,−	0.45 ± 0.16	0.49 ± 0.22
	−,−,−	4.46 ± 0.08	4.86 ± 0.95
	+,+,+	Over10	15.49 ± 5.12

(+): positive, (−): negative.

**Table 3 ijms-20-06260-t003:** Comparison of the method developed in this study with other methods for aflatoxin B1.

Author	Strip Type	Materials	Sample	Limit of Detection	Time	Image	Ref.
Moon et al.	one-dot	40 nm AuNPs	buffer	10 μg/mL	10 min		[41]
Phong et al.	line	AuNPs	buffer	ND	ND		[37]
Delmulle et al.	line	40 nm AuNPs	pig feed	5 μg/kg	10 min		[14]
Liao and Li	line	20 nm silver core AuNPs	rice, wheat, sunflower, cotton, chillies and almond	0.3 ng/mL	10 min		[15]
Shim et al.	Dot aptamer	(Cy5)-modified a single strand-DNA	buffer/corn	0.1/0.3 ng/g	>30 min	Fluorescent apparatus	[18]
Anfossi et al.	line	40 nm AuNPs	maize	1 ng/g	10 min	OpticSlim + Labtop	[17]
Di Nardo	Single line	30 nm and 72 nm AuNPs	maize	2 ng/g AFB1 1 μg/g fumonisins	10 min		[32]
Di Nardo	Single line	30 nm and 75 nm AuNPs	wheat, pasta, pastry	1 and 50 ng/g	10 min	smartphone	[33]

ND: indicates that the Limit of Detection could not be determined.

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
