# Peer review of "Developing Gold Nanoparticles-Conjugated Aflatoxin B1 Antifungal Strips"

_ijms, 2019, doi:10.3390/ijms20246260_

Round 1

Reviewer 1 Report

The authors have satisfied several requests of the first review; however, the text must be further improved.

In order to make your paper more complete about the literature, in the introduction you can cite the general review on LFIAs for mycotoxins of Anfossi et al. “Lateral-flow immunoassays for mycotoxins and phycotoxins: A review” DOI: 10.1007/s00216-012-6033-4.

Moreover, Anfossi et al. developed a quantitative LFIA for the detection of aflatoxins in maize (Development of a quantitative lateral flow immunoassay for the detection of aflatoxins in maize DOI:10.1080/19440049.2010.540763)

About the qualitative detection of AFB1, Di Nardo et al. developed a multiplex LFIA for the detection of AFB1 and fumonisins. (Multicolor immunochromatographic strip test based on gold nanoparticles for the determination of aflatoxin B1 and fumonisins. doi: 10.1007/s00604-017-2121-7)

Finally, about the quantitative detection of AFB1, Di Nardo et al. developed a LFIA for the simultaneous detection of AFB1 and Fumonisins. Qualitative results were obtained just observing the strips, while for quantitative results, the images of the strips were acquired with a smartphone and then analysed through RGB data analysis (Colour-encoded lateral flow immunoassay for the simultaneous detection of aflatoxin B1 and type-B fumonisins in a single Test line. (https://doi.org/10.1016/j.talanta.2018.09.037).

There is still a lot of confusion in the text, for example, line 220: you wrote that you used SEM in order to evaluate the LOD. Moreover, in different sentences you used AFB1 instead of Ab vs AFB1. Please, read more carefully what you wrote Please read more carefully what you wrote, throughout the whole article.

Please, make a table in which you compare the analytical performances of your developed LFIA with the ones already published.

Please, provide images of the Black smartphone accessory (55 × 78 × 30 mm3) and the ICTS cartridge (70 × 18 × 4.8 mm3)

Why your Test and Control lines are so irregular?

Author Response

Dear Reviewer,

Many thanks for taking your time to read carefully of our manuscript. Your suggestions are very helpful . We have taken this opportunity to improve our manuscript. We included four references, one table to compare our method with other works and images of Black smartphone accessory and ICTS cartridge. The manuscript has been sent to proof reading. The responds of each question and manuscript  are in the attachment. The red color high light the changes in the manuscript.

The authors have satisfied several requests of the first review; however, the text must be further improved. In order to make your paper more complete about the literature, in the introduction you can cite the general review on LFIAs for mycotoxins of Anfossi et al. “Lateral-flow immunoassays for mycotoxins and phycotoxins: A review” DOI: 10.1007/s00216-012-6033-4. Moreover, Anfossi et al. developed a quantitative LFIA for the detection of aflatoxins in maize (Development of a quantitative lateral flow immunoassay for the detection of aflatoxins in maize DOI:10.1080/19440049.2010.540763). About the qualitative detection of AFB1, Di Nardo et al. developed a multiplex LFIA for the detection of AFB1 and fumonisins. (Multicolor immunochromatographic strip test based on gold nanoparticles for the determination of aflatoxin B1 and fumonisins. doi: 10.1007/s00604-017-2121-7). Finally, about the quantitative detection of AFB1, Di Nardo et al. developed a LFIA for the simultaneous detection of AFB1 and Fumonisins. Qualitative results were obtained just observing the strips, while for quantitative results, the images of the strips were acquired with a smartphone and then analysed through RGB data analysis (Colour-encoded lateral flow immunoassay for the simultaneous detection of aflatoxin B1 and type-B fumonisins in a single Test line. (https://doi.org/10.1016/j.talanta.2018.09.037).

Respond: All the above manuscripts have been cited. The references have been embedded in introduction and discussion.

Moreover, Anfossi et al. developed a quantitative LFIA for the detection of aflatoxins in maize [16]. A competitive reaction between a biotin-modified aptamer specific to AFB1 and fluorescent cyanine 5-modified DNA probes formed the basis of a dot assay developed by Shim et al. on an LFIA test strip for detection of AFB1 [17]. The fluorescence intensity of the dot was recorded by a fluorescence apparatus coupled to a desktop computer or laptop, which possessed rapid processing speeds and stable performances. However, these bulky and heavy devices limited their wide application in the trend of family and personal care [18-20]. Alternatively, a mobile device-based strip reader could satisfy the requirement of high portability and feature-rich testing. The mobile health market is rapidly developing, and portable diagnostic tools provide an opportunity to increase the availability of health care and decrease costs [21]. Following the developments of various smartphone-based strip readers for quantitative measurements of human diseases [22-30], smartphone analysis for detection of AFL on LFIAs has been also reported earlier this year [31]. The limit of detection (LOD) of gold nano particles (AuNPs) based LFIA has been dramatically improved from 10 μg/mL to 1 ng/g [1-3,12,14,17]. This scenario motivated the development of new strategy providing quantitative analyte concentration for testing LFIAs. So far, the AuNPs sized 30-40 nm have been employed for AFB1 conjugation in literature [1-3,12,14,17]. In order to get a different colour band of LFIAs, Di Nardo et al. has employed blue (desert rose-like, mean diameter ca.75 nm) AuNPs [31,32].  There is a strong association between the AuNPs formulation and colour chang [33,34]. The associated colour can be employed for a number of applications, and therefore, continued refinement of AuNPs synthesis can provide desirable band for LFIAs.

There is still a lot of confusion in the text, for example, line 220: you wrote that you used SEM in order to evaluate the LOD. Moreover, in different sentences you used AFB1 instead of Ab vs AFB1. Please, read more carefully what you wrote throughout the whole article.

Respond:: The phrase has been deleted see line 226. The ‘antibody’ has been added with ‘AFB1’ together.

Please, make a table in which you compare the analytical performances of your developed LFIA with the ones already published.

Respond:: New table 3 has been included.

Table 3. Comparison of the method developed in this study with other methods for aflatoxin B1.

Author

Strip type

Materials

sample

Limit of detection

Time

Image

Reference

Moon et al.

one-dot

40 nm AuNPs

buffer

10 μg/mL

10 mins

[39]

Phong et al.

line

AuNPs

buffer

ND

ND

[36]

Delmulle et al.

line

40 nm AuNPs

pig feed

5 μg/kg

10 mins

[12]

Liao and Li

line

20 nm silver core AuNPs

rice, wheat, sunflower, cotton, chillies and almond

0.3ng/mL

10 mins

[13]

Shim et al.

Dot aptamer

(Cy5)-modified a single strand-DNA

buffer/corn

0.1/0.3ng/g

>30 mins

Fluorescent

apparatus

[17]

Anfossi et al.

line

40 nm AuNPs

maize

1 ng/g

10 mins

OpticSlim +  Labtop

[16]

Di Nardo

Single

line

30 nm and 72 nm AuNPs

maize

2 ng/g AFB1

 1 μg/g fumonisins

10 min

[31]

Di Nardo

Single line

30 nm and 75 nm AuNPs

wheat, pasta, pastry

1 and 50ng/g

10 mins

smartphone

[32]

Please, provide images of the Black smartphone accessory (55 × 78 × 30 mm3) and the ICTS cartridge (70 × 18 × 4.8 mm3)

Respond: New Figure 6S has been included in supplementary materials.

Why your Test and Control lines are so irregular?

Respond: The lateral-flow strips were manually assembled . (see the line 44 in supplementary material )

Best regards,

Furong Tian

Reviewer 2 Report

Authors answered properly the reviewer comments. This improved the overall quality of the manuscript.

Regards

Author Response

Dear Reviewer,

Many thanks for taking your time to read carefully of our manuscript. Your suggestions are very helpful. The manuscript has been sent to proof reading. The changes are at below in red color.

Best regards,

Furong Tian

Round 2

Reviewer 1 Report

The article has been considerably improved. However, before publication, I warmly invite the authors to carefully check the lines 186-188. The authors found "a second-order polynomial relationship between the ratio and AFB1 concentration". However, for competitive immunoassay you should treat your data with a 4-parameter logistic equation.

Please check this point.

Please also check grammar and typing errors.

Author Response

The article has been considerably improved. However, before publication, I warmly invite the authors to carefully check the lines 186-188. The authors found "a second-order polynomial relationship between the ratio and AFB1 concentration". However, for competitive immunoassay you should treat your data with a 4-parameter logistic equation.

Answer: We have employed the 4-parameter logistic equation to our data. The new figure 7 is replaced. The concentration has been recalculated. It looks much better than before. The equation is explained in method and result sections.

Please also check grammar and typing errors.

Answer: The grammar ans typing errors have been checked. 

This manuscript is a resubmission of an earlier submission. The following is a list of the peer review reports and author responses from that submission.

Round 1

Reviewer 1 Report

Review of “Developing gold Nano particles conjugated aflatoxin B1 antifungal strips” for International Journal of Molecular Sciences, as an Article, by Sojinrin et al.

The authors present results on an assay for aflatoxin B1 (AFB1) which is a fungal pathogen in food samples. They use a paper immunochromatographic assay with gold nanoparticles conjugated to an antibody that can bind to AFB1. The assay is competitive, where a loss of color indicates the presence of the toxin in the sample. A paper immunoassay has advantages over other methods of detection, such as HPLC, as it is low-cost and easier to use.

The authors synthesize the AuP-BSA-AFB1 conjugates and characterize them by NMR, SEM, optical spectroscopy, and zeta-potentiometry. They use them to run the test with AFB1 in both pure and food samples, and compare the performance of the assay to HPLC.

There are some issues that need to be addressed before publication. Most notably, there are other papers that have been previously published on AFB1 immunochromatographic strip assays, and so the novelty of the work here is not clearly articulated.

1.) The authors need to properly put this work in context relative to previously published papers on AFB1 strip assays. There are many papers in the literature on lateral flow immunoassay strips for detecting aflatoxins using gold nanoparticle antibody conjugates. For example, these are some references below that also discuss AFB1 strip assays.

-Delmulle et al., J. Agric. Food. Chem 2005 , DOI: 10.1021/jf0404804,

-Jia-Yao Liao, Hang Li DOI 10.1007/s00604-010-0431-0
Microchimica Acta December 2010 (not cited in MS, should be cited)

-Won-Bo Shim Min Jin Kim Hyoyoung Mun Min-Gon Kim
Biosensors and Bioelectronics, Volume 62, 15 December 2014, 288-294.
doi.org/10.1016/j.bios.2014.06.059

Therefore, the authors should clarify how this work is novel with respect to these previous papers and similar ones in the literature. The authors should also cite this previous work, as it is not well represented in the references.

2.) The authors should quantify the limit of detection (LOD) of the strips. They quantify the strip with the lowest concentration that is detectable, but this is not the correct definition of the LOD.

3.) It is well known that immunochromatographic strips can be highly variable in the test line intensity. The author should show triplicates of all the assays and report it with standard deviations, as it will help show the variability of the assay.

4.) In Figure 2, the caption should be labeled properly (it seems to also include the AFB1, and the AFB1 antibody).

5.) The authors use an aggregation index for the SPR shift in Figure 4. There is significant work on using aggregation indices (AIs) in the literature, and these should be referred to and cited here.

6.) The authors should show error bars on Figure 6e for the HPLC data.

7.) SEM is not a useful way to confirm conjugation to AuNPs. Clustering is not statistically significant and can occur simply due to AuNP-substrate interactions, where it is not attributable to the antibody-antigen interactions. This should be emphasized in the manuscript. The authors should confirm conjugation using other techniques, such as dynamic light scattering (DLS), shifts in gel electrophoresis, ELISA, and fluorescence.

8.) Because the authors compare the performance of the immunochromatographic strips with HPLC, it would be good for them to include a side-by-side comparison of all the factors involved in running the assay for HPLC vs. strips (time to run, sample required, cost, etc.).

Reviewer 2 Report

This is an interesting work done by T. Sojinrin et al. who developed gold nanoparticles conjugated aflatoxin B1 antifungal strips.

The conclusions are well supported by results. However, the manuscript needs to be improved in terms of description of the experiments and the used conditions that are not clearly described. The English should be improved. For example, the T line and C line function (Figure 5) was not clear for the reviewer. What is the “coating antigen” ? The description of the process should be re-written in a more simple manner. Moreover, the figures and Tables are not well cited in the manuscript which did not help to understand some of the experiments and results. This should be corrected.

Some information are missing and some others are repeated several times in the manuscript and the supporting information. Missing information should be added and redundancy must be avoided.

The novelty of the developed approach is not clear to the reviewer. Several studies on the same topic using conjugated gold nanoparticles to detect aflatoxin can be found in the literature. These articles should be discussed in the manuscript and the novelty of the herein work needs thus to be demonstrated.

Moreover, the quality / robustness in terms of reliability of the fabrication of the POC of the strips was not addressed. Is this mutli-step process to obtain a strip reproducible? This should be clearly specified and discussed with data in support. What are the storage conditions of the strips as well as their stability ? How many times the strip can be used ?

Authors based their assay on color change due to aggregation. Is aggregation a repeatable process ? From Figure 4 it seems that solutions are still homogeneous despite the aggregation. How can this be explained ?

Reference 21 needs to be completed.

From the scale specified in Figure 3, it seems that authors were working with microparticules and not nanoparticules. What is the correct scale ?

Regards

Reviewer 3 Report

The paper of Tobiloba Sojinrin et al. try to describe the development of a Lateral Flow Immunoassay to detect Aflatoxin B1.

There are plenty of Lateral Flow Assay tests to detect AFB1 in the literature, and this paper does not add anything new. Moreover, the authors do not even mention one of the works already published in this regard.

There is a lot o confusion in the text. Sometimes the authors refer to AFB1 antigen, when it is clear that the subject would be the anti-AFB1 antibody.

Please do not use the adjective “conversational” referred to HPLC.

Different terms are used incorrectly (such as hapten).

Even in the introduction there are incorrect references to the literature. It is certainly not at a 2015 review that reference should be made to say that the main components of a lateral flow assay strip are sample pad, conjugate pad, nitrocellulose membrane and absorbent pad.

No information about sample preparation for LFIA analysis are given (are the steps the same as for HPLC analysis?), neither on the volume used to perform the analysis

No information about the instrumentation to depose T and C lines on the nitrocellulose membrane are given. Neither on the rate/amount of deposition. Which nitrocellulose membrane did you use?

Just some notes:

It is really strange tha Gold colloid suspensions at size at 10 nm (as reported in “materials and methods” showed a SPR band at 530 nm (as reported in “Confirmation of Physical Characteristic of AuNPs”)

Lines 143 to 153 in the main text are the same of lines 29 to 40 in Supplementary materials, adding no useful information for the reader.

The LOD of the LFIA must be determined in another way.

In line 169 the Authors referred to Supplementary materials for the cut-off definition, but no information are given in the Supplementary materials about that.

Lines 175 to 177: it is exactly the opposite.

Line 287: The title is “Synthesis of AFB1-BSA gold nanoparticles” and than you wrote about AFB1 antibody.

Supplementary materials

In “Preparation of Sample and Conjugate pad strips” no information about the conjugate pad preparation are given.

Why did you soak the sample pad in ethanol?

Line 25: “The AFB1 antibody was conjugated with anti-mouse antibody.” it does not make any sense.

Figure S2-5: I cannot see very well the conjugate pad

Please provide images of the strips after the analysis with a better resolution